# Essential Variables for Environmental Monitoring: What Are the Possible Contributions of Earth Observation Data Cubes?

**Gregory Giuliani** [1,2,*] **, Elvire Egger** [2] **, Julie Italiano** [2] **, Charlotte Poussin** [2] **, Jean-Philippe Richard** [2] **and Bruno Chatenoux** [2]

1   Institute for Environmental Sciences, University of Geneva, enviroSPACE, Bd Carl-Vogt 66, CH-1205 Geneva, Switzerland

2   Institute for Environmental Sciences, University of Geneva, GRID-Geneva, Bd Carl-Vogt 66, CH-1211 Geneva, Switzerland; Elvire.Egger@etu.unige.ch (E.E.); julie.italiano@gmail.com (J.I.); Charlotte.Poussin@unige.ch (C.P.); jean-philippe.richard@unepgrid.ch (J.-P.R.); bruno.chatenoux@unepgrid.ch (B.C.)

*   Correspondence: gregory.giuliani@unige.ch; Tel.: +41-(0)22-379-07-09

**Abstract:** Environmental sustainability is nowadays a major global issue that requires efficient and effective responses from governments. Essential variables (EV) have emerged in different scientific communities as a means to characterize and follow environmental changes through a set of measurements required to support policy evidence. To help track these changes, our planet has been under continuous observation from satellites since 1972. Currently, petabytes of satellite Earth observation (EO) data are freely available. However, the full information potential of EO data has not been yet realized because many big data challenges and complexity barriers hinder their effective use. Consequently, facilitating the production of EVs using the wealth of satellite EO data can be beneficial for environmental monitoring systems. In response to this issue, a comprehensive list of EVs that can take advantage of consistent time-series satellite data has been derived. In addition, a set of use-cases, using an Earth Observation Data Cube (EODC) to process large volumes of satellite data, have been implemented to demonstrate the practical applicability of EODC to produce EVs. The proposed approach has been successfully tested showing that EODC can facilitate the production of EVs at different scales and benefiting from the spatial and temporal dimension of satellite EO data for enhanced environmental monitoring.

**Keywords:** essential variables; climate; biodiversity; water; earth observations; data cube

---

## 1. Introduction

The latest Global Risks Report from the World Economic Forum (WEF) identified for the first time in its 10-year outlook survey that the top five global risks in terms of likelihood and impact were all related to the environment, ranking climate change and biodiversity loss as major global threats [1]. Therefore, environmental sustainability is recognized as a critical global issue that demands achieving a good balance between the exploitation of natural resources for socio-economic development, and conserving ecosystem services that are critical to everyone's wellbeing and livelihoods [2].

To reach this objective, it is essential to provide accountable scientific knowledge to policy- and decision-makers to enable them to develop effective policies based on evidence [3,4]. However, this is still a challenging task because this requires organizing actions (i.e., from data acquisition to knowledge generation) in coherent and coordinated workflows. These data value chains are a series of transformations to produce new insights from data [5] that can follow the data-information-knowledge-wisdom pattern

to provide evidence about the limits of our planet [6,7]. This requires a paradigm shift from data-centric to more knowledge-centric approaches [8] integrating different datasets from various disciplines both environmental and socio-economic conditions [9]. Ultimately, this allows describing and assessing environmental conditions at different scales (e.g., national, regional, global) and understanding the interactions between the different components of the Earth system (e.g., atmosphere, hydrosphere, biosphere, geosphere) and possibly predicting future changes [10]. However, such a knowledge-centric approach has not yet been widely applied for environmental monitoring [11,12].

The concept of essential variables (EV) has emerged in various scientific communities as a means to adequately describe and monitor the evolution of Earth system components (e.g., atmosphere, biosphere, hydrosphere) through a set of observational data. These data should be (1) relevant (i.e., critical for characterizing a system and its changes); (2) feasible (i.e., methodologies are scientifically sound, technically feasible); (3) cost-effective (i.e., low cost of implementation, maintenance, archiving and (4) support policy frameworks [13,14]. EVs can be understood as *"a minimal set of variables that determine the system's state and developments, are crucial for predicting system developments, and allow us to define metrics that measure the trajectory of the system"* [15]. Currently, EVs are already been defined for climate [16–18], biodiversity [19,20], water [21], oceans [22] and other domains have started the process such as agriculture [23], ecosystems [24], mountains, extractives [25], air quality [26] or renewable energies [27] as well as socio-economic [13,28]. Timely and reliable access to EVs can provide the basis for accountable scientific understanding and knowledge on environmental conditions to enhance informed and evidence-based policy process [29].

To facilitate environmental monitoring, our planet has been under continuous observation from satellites since 1972, providing synoptic, regular, multi-spectral observations of the planet. These Earth observation (EO) data can provide the long baseline required to determine trends, define present conditions, and inform future evolution [30]. EO data have the potential to drive progress in key national and international development agendas, providing new insights and supporting better policy development across diverse environmental sustainability issues [31,32]. Indeed, governments have national and international commitments and obligations as well as national environmental programs. They all need information that is synoptic, consistent, spatially explicit and sufficiently detailed to capture anthropogenic impacts. However, this vision of EO data-driven decision making has not been fully addressed and challenges remain [33]. The full information potential of EO data has not yet been realized, because many big data challenges (e.g., volume, variety, velocity), related complexity barriers (e.g., data preparation, model integration and prediction) and lack of capacities/skills hinder effective and efficient use of EO data in policy and decision-making processes [34,35].

Traditional approaches to the acquisition, management, distribution, and analysis of satellite EO data have limitations (e.g., data size, heterogeneity, complexity) that impede their massive use and analysis [36]. To address these issues and leverage the information potential of EO data, EO Data Cubes (EODC) have been developed to strengthen the connection between data, applications and users [37]. EODC are tackling Big Data challenges by providing efficient and effective solutions to store, organize, manage and analyze large volumes of freely available data in analysis ready (ARD) format [38,39]. Various implementations exist mostly at the national and regional scales such as in Australia [40,41], Africa [42], Armenia [43], Austria [44], Brazil [45], Colombia [46], Mexico [31], Switzerland [47] and Taiwan [48]. However, none of these EODC are neither explored the potential nor routinely generating EVs.

Consequently, the aim of this paper is to (1) explore and define which EVs can be generated using EODC in the three most advanced domains: climate, biodiversity and water; and (2) demonstrate the applicability of EODC in generating selected EVs at local and national scales. This can help understanding how EODC can contribute and strengthen existing environmental monitoring systems with the regular provision of EVs.

## 2. Methodology and Implementation

The definition and development of EVs in various thematic and scientific domains usually follows a community process to reach an agreement on a set of observables/measurements that can be considered as essential and that satisfy the needs of national to global monitoring, reporting, research or forecasting [15]. Among the different scientific communities, climate, biodiversity and water are those that can be considered the most mature domains and where EVs definition process is the most advanced [24]. For each of these three domains, EVs are reviewed and evaluated according to their level of measurability using remotely-sensed data time-series. They are then classified following three labels: (Y)es-(P)atrial-(N)o. Yes, means that an EV is fully measurable using remote-sensing data time-series; Partial relates to EVs that need additional information (e.g., in-situ data, models); No, indicates that they are not measurable using remote-sensing. In addition, main spatial ((L)ocal-(N)ational-(R)egional-(G)lobal) and temporal ((H)our-(D)ay-(W)eek-(M)onth-(S)eason-(A)nnual) scales for monitoring are also considered. Classification choices are made based on literature search in relevant scientific libraries such as Science Direct, Scopus, Web of Science or Google Scholar.

### 2.1. Essential Climate Variables

Originally, the concept of EVs has been defined by the climate community through the effort led by the Global Climate Observing System (GCOS) which established a set of 52 essential climate variables (ECV). These ECVs are supposed to support the work of the United Nations Framework Convention on Climate Change (UNFCCC) and the Intergovernmental Panel on Climate Change (IPCC) [17]. They were selected for their relevance to characterize the climate system as well as their technical and economic feasibility for systematic observations from satellite remote-sensing or in-situ measurements [16,49]. ECVs are generated by different organizations (e.g., European Space Agency, National Oceanic and Atmospheric Administration) and can be searched and accessed through the Global Observing Systems Information Center (GOSIC) that provides a single entry-point to ensure a harmonized and integrated search across various data holdings [50]. However, from a user-perspective accessing ECVs is still difficult because of the variety of data formats, access methodologies, spatial coordinates and temporal resolution [18] making them difficult to integrate with other datasets [51,52].

The full list of ECVs (Table 1) has been extracted from the GCOS website (https://gcos.wmo.int/en/essential-climate-variables).

**Table 1.** Essential climate variables (ECV) and possible contributions from remote sensing. Y-Yes, P-Patrial, N-No; L-N-R-G, ((L)ocal-(N)ational-(R)egional-(G)lobal); H-D-W-M-S-A, ((H)our-(D)ay-(W)eek-(M)onth-(S)eason-(A)nnual).

| | Class | ECV | Remote Sensing | Spatial Scale | Temporal Scale |
|---|---|---|---|---|---|
| Atmosphere | Surface | Precipitation | Y [53] | L-N-R-G | H-D-W-M-S-A |
| | | Pressure (surface) | N | L-N-R-G | H-D-W-M-S-A |
| | | Surface Radiation Budget | Y [54] | R-G | H-D-W-M-S-A |
| | | Surface Wind Speed and direction | Y [55] | L-N-R-G | H-D-W-M-S-A |
| | | Temperature (near surface) | Y [56] | L-N-R-G | H-D-W-M-S-A |
| | | Water Vapor (surface) | Y [55] | R-G | H-D-W-M-S-A |
| | Upper Atmosphere | Earth Radiation Budget | Y [57] | G | W-M-S-A |
| | | Lighting | Y [58] | G | W-M-S-A |
| | | Temperature (upper air) | P [59] | R-G | H-D-W-M-S-A |
| | | Water Vapor (upper air) | P [60] | R-G | H-D-W-M-S-A |
| | | Cloud properties | P [61] | R-G | H-D-W-M-S-A |
| | | Wind speed and direction (upper air) | Y [55] | L-N-R-G | H-D-W-M-S-A |
| | Atmospheric Composition | Aerosols properties | P [62] | L-N-R-G | H-D-W-M-S-A |
| | | Carbon Dioxide, Methane and other Greenhouse gases | P [63] | L-N-R-G | H-D-W-M-S-A |
| | | Ozone | Y [64] | L-N-R-G | H-D-W-M-S-A |
| | | Precursors (supporting the Aerosol and Ozone ECVs) | P [65] | L-N-R-G | H-D-W-M-S-A |

**Table 1.** *Cont.*

| | Class | ECV | Remote Sensing | Spatial Scale | Temporal Scale |
|---|---|---|---|---|---|
| **Land** | Hydrosphere | River Discharge | N | L-N-R-G | H-D-W-M-S-A |
| | | Groundwater | P [66] | R-G | H-D-W-M-S-A |
| | | Lakes | Y [67] | L-N-R-G | H-D-W-M-S-A |
| | | Soil Moisture | P [68] | L-N-R-G | H-D-W-M-S-A |
| | Cryosphere | Snow | Y [69] | L-N-R-G | H-D-W-M-S-A |
| | | Glaciers | Y [70] | L-N-R-G | M-S-A |
| | | Ice Sheets and ice shelves | P [70] | L-N-R-G | M-S-A |
| | | Permafrost | P [71] | L-N-R-G | M-S-A |
| | Biosphere | Albedo | Y [72] | R-G | H-D-W-M-S-A |
| | | Land cover | Y [73] | L-N-R-G | A |
| | | Fraction of Absorbed Photosynthetically Active Radiation (FAPAR) | P [74] | L-N-R-G | W-M-S-A |
| | | Leaf Area Index (LAI) | P [74] | L-N-R-G | W-M-S-A |
| | | Above-ground biomass | P [75] | L-N-R-G | M-S-A |
| | | Soil Carbon | P [76] | L-N-R-G | M-S-A |
| | | Land Surface Temperature | Y [77] | L-N-R-G | H-D-W-M-S-A |
| | | Fire | Y [78] | L-N-R-G | H-D-W-M-S-A |
| | | Evaporation from land | P [79] | L-N-R-G | H-D-W-M-S-A |
| | Anthroposphere | Anthropogenic Greenhouse Gas Fluxes | N | L-N-R-G | H-D-W-M-S-A |
| | | Anthropogenic Water Use | N | L-N-R-G | H-D-W-M-S-A |
| **Ocean** | Physical | Ocean Surface Heat Flux | P [80] | R-G | H-D-W-M-S-A |
| | | Sea Ice | Y [81] | R-G | W-M-S-A |
| | | Sea Level | P [82] | R-G | H-D-W-M-S-A |
| | | Sea State | N | R-G | H-D-W-M-S-A |
| | | Sea Surface Salinity | Y [83] | R-G | H-D-W-M-S-A |
| | | Sea Surface Temperature | Y [84] | R-G | H-D-W-M-S-A |
| | | Subsurface Currents | N | R-G | H-D-W-M-S-A |
| | | Surface Currents | P [85] | R-G | H-D-W-M-S-A |
| | | Surface Stress | N | R-G | H-D-W-M-S-A |
| | Biogeochemical | Inorganic Carbon | N | R-G | W-M-S-A |
| | | Nitrous Oxide | N | R-G | W-M-S-A |
| | | Nutrients | P [86] | R-G | D-W-M-S-A |
| | | Ocean Colour | Y [87] | R-G | D-W-M-S-A |
| | | Oxygen | N | L-N-R-G | H-D-W-M-S-A |
| | | Transient Tracers | N | L-N-R-G | H-D-W-M-S-A |
| | Biological/Ecosystems | Marine Habitats Properties | P [88] | R-G | M-S-A |
| | | Plankton | P [89] | L-N-R-G | H-D-W-M-S-A |

## 2.2. Essential Biodiversity Variables

Similarly, the biodiversity community has established essential biodiversity variables (EBV) as a set of variables that are between primary observations and indicators [19,20]. Currently, there are 21 EBVs organized in 6 classes (e.g., genetic composition, species population, species traits, community composition, ecosystem function, ecosystem structure) [90]. These variables are meant to study, report and manage biodiversity change, status and trends [91]. They designed to support global initiatives such as the United Nations Convention on Biological Diversity (CBD) and the Intergovernmental Panel for Biodiversity and Ecosystem Services (IPBES) [92] as well as national reporting commitments [93,94]. Ideally, these variables should benefit from a synergetic use of remote sensing and in-situ measurements [95,96]. However, to access EBV, interoperability challenges are similar to those identified for ECVs [92].

The EBV list (Table 2) has been extracted from the Group on Earth Observations (GEO) Biodiversity Observation Network (BON) website (https://geobon.org/ebvs/what-are-ebvs/) which is providing access to EBVs through its portal (https://portal.geobon.org). The classification is based on the work done by Pettorelli et al. (2016) [95] and Randin et al. (2020) [97] how carefully reviewed the potential contributions of remote sensing for biodiversity.

**Table 2.** Essential biodiversity variables (EBV) and possible contributions from remote sensing.

| Class | EBV | Remote Sensing | Spatial Scale | Temporal Scale |
|---|---|---|---|---|
| Genetic composition | Co-ancestry | N | L | M-S-A |
| | Allelic diversity | N | L-N-R-G | M-S-A |
| | Population genetic differentiation | P | L-N-R-G | M-S-A |
| | Breed and variety diversity | N | L-N-R-G | M-S-A |
| Species population | Species distribution | P | L-N-R-G | M-S-A |
| | Population abundance | P | L-N-R-G | M-S-A |
| | Population structure by age/size class | P | L-N-R-G | M-S-A |
| Species traits | Phenology | P | L-N-R-G | W-M-S-A |
| | Morphology | P | L-N-R-G | M-S-A |
| | Reproduction | N | L-N-R-G | W-M-S-A |
| | Physiology | N | L-N-R-G | W-M-S-A |
| | Movement | N | L-N-R-G | W-M-S-A |
| Community composition | Taxonomic diversity | P | L-N-R-G | M-S-A |
| | Species interactions | N | L-N-R-G | M-S-A |
| Ecosystem functions | Net primary productivity | P | L-N-R-G | H-D-W-M-S-A |
| | Secondary productivity | N | L-N-R-G | H-D-W-M-S-A |
| | Nutrient retention | N | L-N-R-G | M-S-A |
| | Disturbance regime | Y | L-N-R-G | M-S-A |
| Ecosystem structure | Habitat structure | P | L-N-R-G | M-S-A |
| | Ecosystem extent and fragmentation | P | L-N-R-G | M-S-A |
| | Ecosystem composition by functional type | P | L-N-R-G | M-S-A |

## 2.3. Essential Water Variables

Essential water variables (EWV) have emerged from different activities on water-related to the GEO and led by the Integrated Global Water Cycle Community of Practice [98]. Through a community reviewing process to ensure wide acceptance by the community, a set of 16 variables that are linked to applications and end-users [21] and are aiming to support international bodies to provide a comprehensive list of parameters targeting monitoring, modeling, and inter-disciplinary decision support systems [99]. This list has been derived from user-needs/requirements assessment in all GEO user sectors (e.g., agriculture, health, ecosystems, biodiversity, climate, energy, weather).

EWV are listed in Table 3 and extracted from the Global Earth Observation System of Systems (GEOSS) water strategy report [98].

**Table 3.** Essential water variables (EWV) and possible contributions from remote sensing.

| Class | EWV | Remote Sensing | Spatial Scale | Temporal Scale |
|---|---|---|---|---|
| Primary | Precipitation | Y [53] | L-N-R-G | H-D-W-M-S-A |
| | Evaporation and Evapotranspiration | P [79] | L-N-R-G | H-D-W-M-S-A |
| | Snow Cover (and Depth, Freeze Thaw Margins) | P [100] | L-N-R-G | H-D-W-M-S-A |
| | Soil moisture/temperature | Y [56] | L-N-R-G | H-D-W-M-S-A |
| | Groundwater | P [101] | R-G | H-D-W-M-S-A |
| | Runoff/Streamflow/River Discharge | N | L-N-R-G | H-D-W-M-S-A |
| | Lakes/Reservoir levels and Aquifer volumetric change | P [67] | L-N-R-G | H-D-W-M-S-A |
| | Water Quality | P [102] | L-N-R-G | H-D-W-M-S-A |
| | Water use/demand | N | L-N-R-G | D-W-M-S-A |
| | Glaciers/ice sheets | Y [70] | R-G | M-S-A |
| Supplementary | Surface meteorology | P [55] | L-N-R-G | H-D-W-M-S-A |
| | Surface and atmospheric radiation budget | P [57] | R-G | H-D-W-M-S-A |
| | Cloud and aerosols | P [103] | L-N-R-G | H-D-W-M-S-A |
| | Land Cover and vegetation/Land use | Y [73] | L-N-R-G | M-S-A |
| | Permafrost | P [71] | L-N-R-G | W-M-S-A |
| | Elevation/topography and geological stratification | P [104] | L-N-R-G | A |

## 2.4. Implementation

To demonstrate the practical capability to generate EVs, an operational EODC implementation has been used enabling a country to benefit from satellite EO data for environmental monitoring.

Switzerland is facing important environmental challenges because it is a small country with increasing pressures on land resources [105] and it is among the first countries in the world to have an operational EODC providing access to a unique EO analysis-ready data (ARD) archive over the entire national territory [106]. The Swiss Data Cube (SDC) is a tera-scale analytical cloud-based platform offering access to 36 years (e.g., 1984 to present days) of satellite data from Landsat 5-7-8 [47], Sentinel-1 [107], and Sentinel-2 [108] sensors. The SDC facilitates national-scale analyses of large volume of spatially aligned and consistently calibrated satellite EO data. It is an initiative implemented and operated by the United Nations Environment Programme (UNEP)/Global Resource Information Database (GRID)-Geneva in partnership with the University of Geneva, the University of Zurich, and the Swiss Federal Institute for Forest, Snow and Landscape Research (WSL). The SDC is supported by the Federal Office for the Environment (FOEN) and is aiming to contribute to its environmental and reporting mandate while at the same time enabling any Swiss institutions to benefit from the information power of satellite EO data. The archive is updated on a daily basis and contains approximately 12,500 scenes corresponding to a total volume of 5 TB and more than 1000 billion observations/pixels. It is built on the Open Data Cube (ODC), an open-source geospatial data management and analysis software project, led by the Committee on Earth Observation Satellites (CEOS) in partnership with Geoscience Australia, the Commonwealth Scientific and Industrial Research Organisation (CSIRO), the National Aeronautics and Space Administration (NASA), the United States Geological Survey (USGS), Catapult Satellite Applications, and Analytical Mechanics Associates (AMA) [109]. In the backend, PostgreSQL database is used to index data and their metadata; data can be either stored as Network Common Data Format (NetCDF) or Cloud-optimized Geotiff; and data are handled using Geospatial Data Abstraction Library (GDAL) and specific Python libraries (e.g., numpy, proj, matplotlib). The SDC has already provided insightful results for forest monitoring [110], land degradation [111] and snow evolution [112].

To efficiently and effectively process these large volumes of data and developed tailored applications, the SDC provides a Python application programming interface (API) that enables users to write their own processing algorithms [113]. This API is accessible through Jupyter Notebooks, an interactive web-based programming interface that can be used for combining software code, algorithm output and explanatory text [114]. Consequently, using these capabilities, dedicated analysis workflows have been implemented as Python scripts to produce the EVs presented in the next section. Results are then published into a GeoServer instance (https://geoserver.swissdatacube.org/) and properly documented with metadata descriptions and stored in a GeoNetwork catalog (https://geonetwork.swissdatacube.org/geonetwork/srv/eng/catalog.search#/home). This allows an interoperable discovery and access to data products using standardized interfaces such as the Web Map Service (WMS), Web Coverage Service (WCS) and Catalog Service for the Web (CSW). Ultimately, these data can be examined in a web-based application enabling users to visualize, query, and download time-series products generated with the SDC (http://www.swissdatacube.org/viewer).

## 3. Results

Based on the evaluation of ECV, EBV and EBW, approximately 67 from a total number of 89 variables can be fully or partially measured using remote sensing. This corresponds to a percentage of 75%. By domain, this corresponds to 79% of ECV (41 of 52), 57% of EBV (12 of 21) and 88% of EWV (14 of 16). This shows that remote sensing can provide a substantial contribution to measure this EVs.

To test and validate the technical feasibility, identify possible limitations and determine the potential of generating EVs using an EODC, different use-cases have been selected. Climate change, biodiversity loss and water management are among the most important pressures that Switzerland has to overcome in the next future [115]. Based on this consideration and the review done in Section 2, the following essential variables have been selected: (1) for ECV: Soil moisture and Snow cover; (2) for EBV: Ecosystem structure; and finally, for EWV: Lake level and Water quality.

Switzerland is recognized as the water castle of Europe because approximately 6% of Europe's freshwater reserves are stored in the country [116]. Therefore, adequate policies and management

practices are required to preserve and protect this resource. One famous feature of the Swiss landscape is the mountainous regions covered by snow in the winter season. Besides its touristic aspect [117], snow is an important water resource and component of the water cycle, storing water in winter and releasing it in spring during the melting season. Snow cover is an ECV that is important to monitor as it provided useful indications on climate change as well as for effective natural resource management (e.g., flood risk management, water supply) [118]. The contribution of snow melt to water bodies will likely decrease by about 25% by the end of the century and consequently greatly affect the water regime of some of the major European rivers (e.g., Rhône, Rhine and Danube) [119,120]. Detailed information on snow cover evolution can be produced to map snow cover extension using optical sensors data available in the Swiss Data Cube. The Snow Observations from Space (SOfS) algorithm [121] allows monitoring of snow cover evolution and variability over the entire Swiss territory using a time-weighted approach of the Normalized Difference Snow Index [100,122] to compute frequencies of observations. Initial results using 20 years of Landsat observations have evidenced an apparent decrease of snow cover during the winter season [31,112]. The area where permanent snow coverage is observed during the winter season has decreased by about 4% while during the same period the area with little or no snow has increased by about 8% (Figure 1).

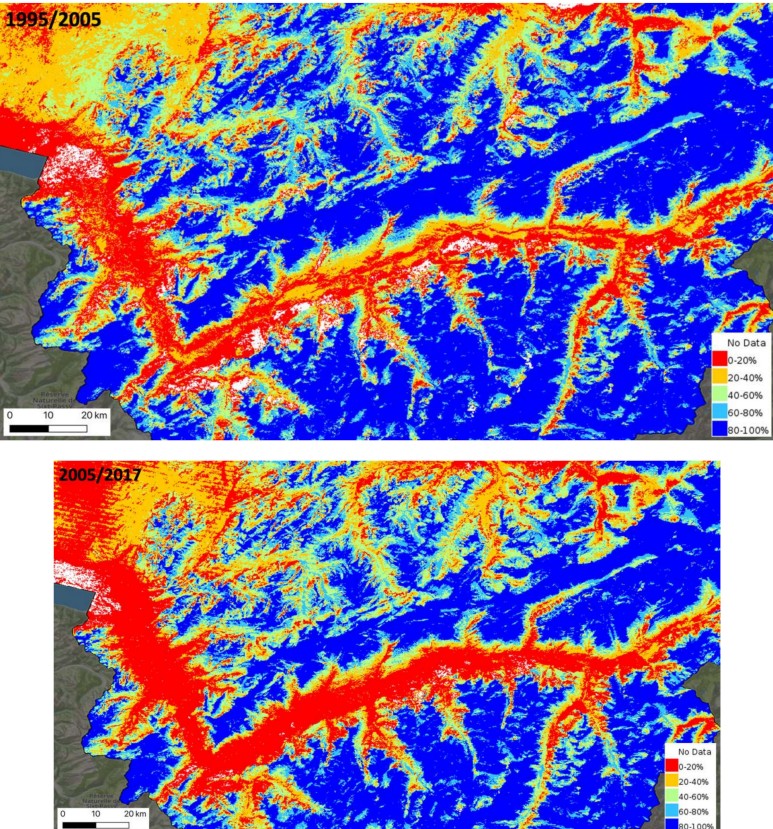

**Figure 1.** Comparison of snow observations frequencies between the periods 1995/2005 (**above**) and 2005/2017 (**below**). Areas with permanent snow cover over the winter season (dark blue) have significantly decreased whereas at the same time areas with little or no snow (red) have increased. Data source: Landsat.

Such results can be an indication of changing climate conditions over the Swiss Alps and can help follow the evolution trend of snow cover in this Alpine region.

Another ECV that can be useful to provide useful information on water conditions is the Soil moisture. It can be measured in various ways but the Normalized Difference Water Index (NDWI) is recognized as a good proxy to estimate water content in vegetation and soils [123]. The NDWI

estimates soil moisture and canopy water content using the reflecting properties in the near-infrared (NIR) and short-wave infrared (SWIR) bands and is calculated as follows [124]:

$$NDWI = (NIR - SWIR)/(NIR + SWIR), \tag{1}$$

Usually, NDWI is a function of local climate conditions and soil properties controlling water availability [125]. NDWI has demonstrated its usefulness in different circumstances such as drought monitoring, early warning, crop monitoring and soil moisture in grasslands and shrublands [126–128]. In the Swiss Data Cube, a Python script has been implemented to produce different satellite-derived indices allowing users to easily perform time-series analysis of annual, seasonal or monthly mean. Annual and seasonal NDWI means (with their respective standard deviations) have been computed for the entire country from 1984 to 2019 using Landsat data. Preliminary results allow researchers to explore both spatial and temporal variation of NDWI across Switzerland (Figure 2).

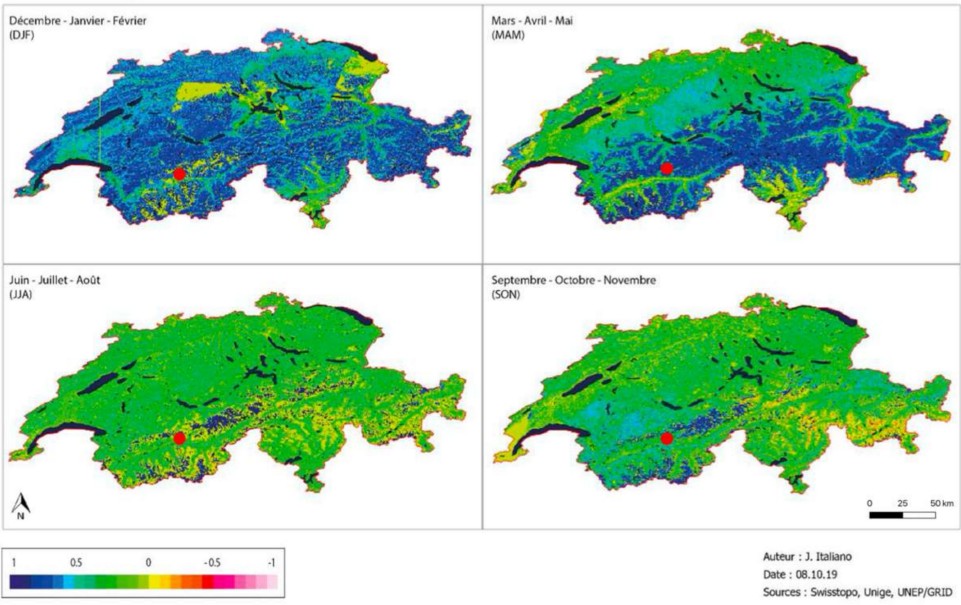

**Figure 2.** Normalized Difference Water Index (NDWI) seasonal mean (Year 2019) from Winter (**upper-left**), Spring (**upper-right**), Summer (**lower-left**) and Autumn (**lower-right**). These maps show the evolution over a year of the water content in vegetation and soils.

Such datasets allow soil moisture over a dedicated area to be explored. For example, users can select a protected area in the Valais canton named "Bois de Finges" and compute a zonal statistic over the annual mean time-series (Figure 3). This gives a curve that indicates that soil moisture is relatively stable over time, even if a small decrease can be observed, but more interestingly shows various important declines corresponding to major drought events that occurred in 1986, 2003 and 2018. This demonstrates the potential of NDWI to monitor soil moisture and drought conditions.

Still concerning water, lakes are also a major feature of the Swiss landscape having more than 100 lakes of a surface larger than 0.3 square kilometers [115]. Most of these lakes are artificially regulated and, therefore, their levels can be generally considered as stable [129,130]. However, in the last years, Switzerland has suffered regularly from severe drought conditions that have affected small natural and unregulated lakes showing an important decrease in their respective water levels. Water level, being an EWV, can be evaluated using satellite imagery using both optical and/or radar imagery [131]. In the French-speaking part of Switzerland, two lakes are regularly mentioned in the news showing a substantial decrease in their respective water levels during the summer season. These are the lake des Brenets, located in the north-western part at the Swiss/French border, and the lake Bret, on the eastside of Lausanne city. Using the Water Observations from Space algorithm (WOfS) [132] it is possible to detect surface water and evaluate the frequency of whether water is permanent or temporary for a

given pixel through over a dedicated time-frame. Water percentages have been computed for the 2017 and 2018 summer months using Sentinel-2 data for both lakes (Figures 4 and 5).

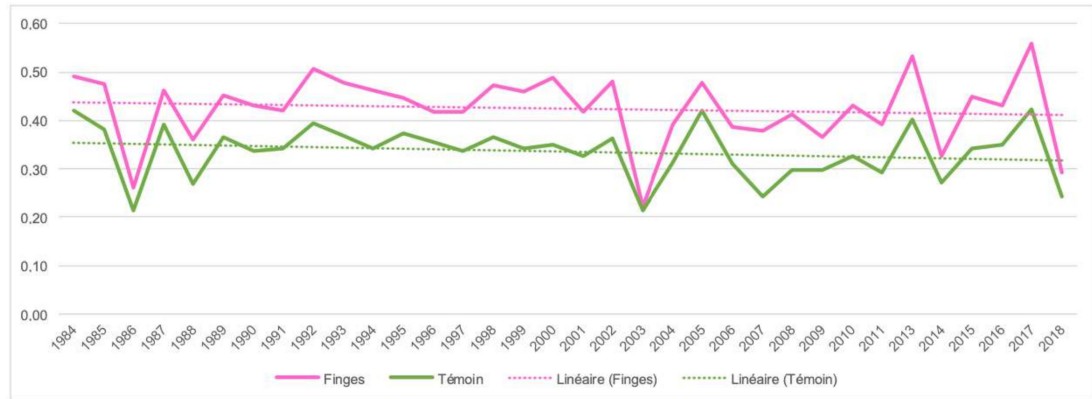

**Figure 3.** NDWI Annual mean time-series for the Bois de Finges protected area showing important decrease in water content corresponding to major drought events.

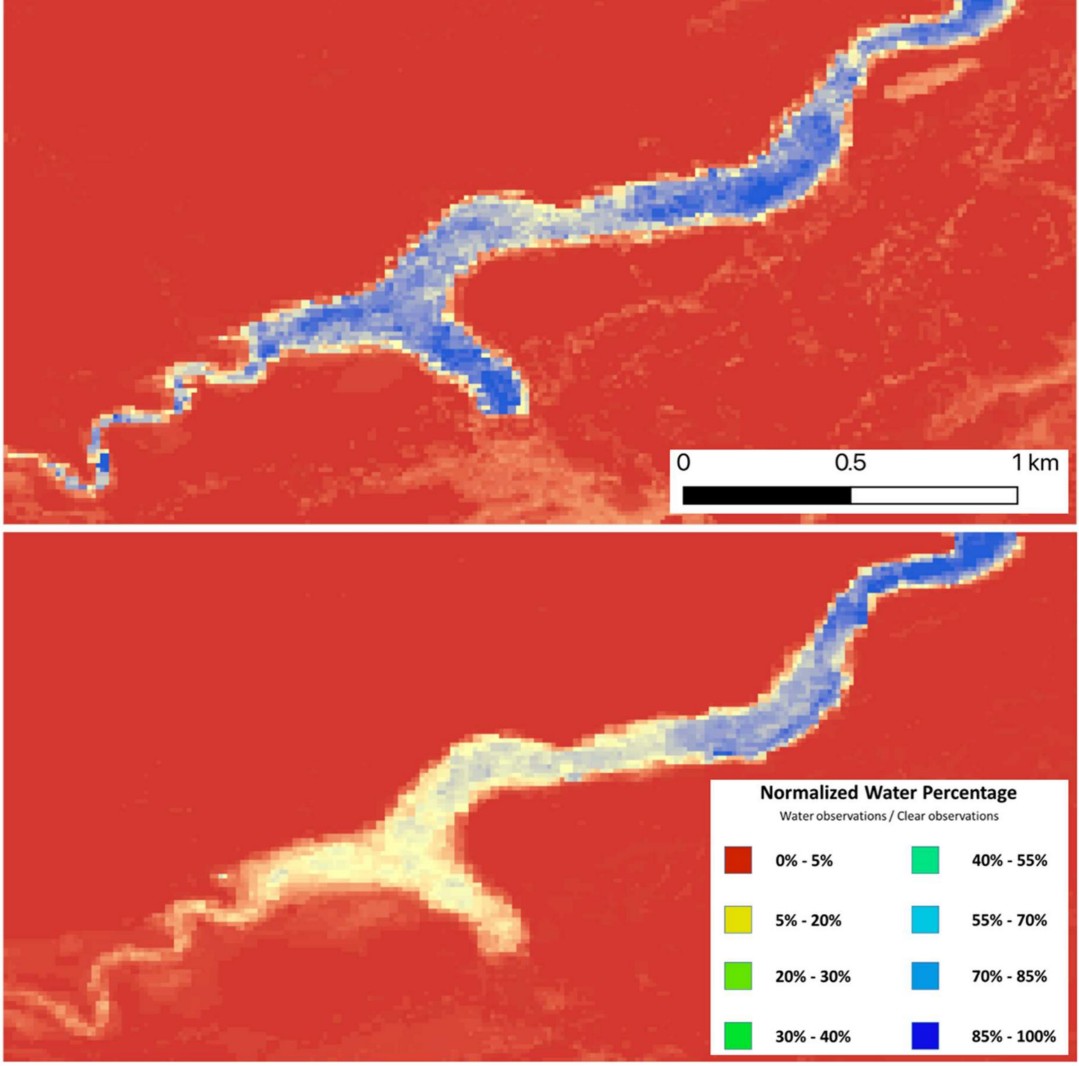

**Figure 4.** Water percentages for the lake des Brenets in 2017 (**above**) and 2018 (**below**). Water level severely decreased in 2018 showing that the east side of the lake was covered approximately only 50% of the time (in yellow) compared to permanent water (blue) and no water (red). Data source: Sentinel-2.

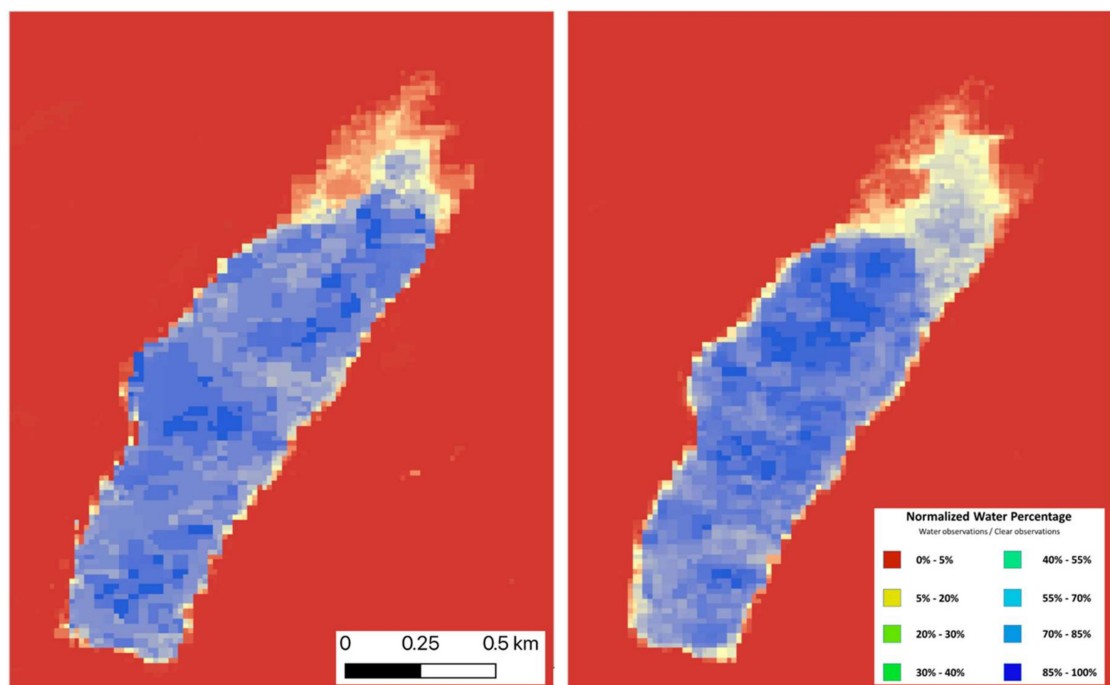

**Figure 5.** Water percentages for the lake de Bret in 2017 (**left**) and 2018 (**right**). The water level shows a similar pattern with the north-west side of the lake evidencing a clear decrease in water level. Data source: Sentinel-2.

This shows that for the same period of the year, the 2018 drought strongly affected the two lakes and a clear decrease in water levels can be observed. Large areas have been completely drained, severely impacting leisure activities such as sailing or fishing.

Remote-sensing techniques have been extensively applied for Water quality assessments on oceans, seas, and lakes and can help to estimate and monitor mineral water quality indicators such as turbidity, chlorophyll content, surface temperature or water colors [87,102,133]. Total suspended matter (TSM) allows the average amount of suspended matter in a water body to be determined and, therefore, can be considered as a relatively good proxy to estimate water quality. It can help to evidence areas of high turbidity that can have various causes such as pollution, vegetation or mineral induced turbidity. Lakes Thun and Brienz are located in the central part of Switzerland at the foothill of the Alps. These two lakes are feed only by the Aar river, a tributary of the Rhine, that takes its source in the Aare Glacier in the Bernese Alps. Consequently, the Aar river is carrying an important sediment load that can somehow affect the water quality of these two lakes [134]. The TSM algorithm implemented by the ODC can be used with Landsat data from April to August 2016 to compute the average value per pixel of suspended matter (Figure 6).

Results show that most of the sediment load is deposited in the Brienz lake whereas the Thun lake appears clearer having virtually no sediment deposited in the lake. This result is confirmed also with a visual inspection of a true-color composite (Figure 7) for the same period that shows lake Brienz having a color that is desaturated and is whiter than the Thun lake that has a better water quality.

The last example relates to EBV and how ecosystem structure can be evaluated. By structure, it is implied the set of biophysical properties of ecosystems that create the conditions of the structural components of ecosystems that lead to and maintain biodiversity characteristics [19]. Ecosystem extent and fragmentation can be estimated using satellite data. To quantify the spatial extent of vegetation, one can use the fractional cover (FC) methodology. The fraction of vegetation cover can be derived using a linear unmixing methodology to evaluate the proportion of photosynthetic vegetation (PV), non-photosynthetic vegetation (NPV) and bare soil (BS) within each pixel and codified in red/green/blue colors [135]. In Figure 8, Landsat 8 data has been used to compute FC for estimating the spatial extent

of vegetation in a portion of the Swiss low-land in the Greater Geneva area during the summer of 2003. This map shows that most of the area is covered by photosynthetic vegetation and can help to better determine the structure of vegetated areas as well as assessing the impacts of drought that caused the death of vegetation during the heatwave an appearing as NPV on the map.

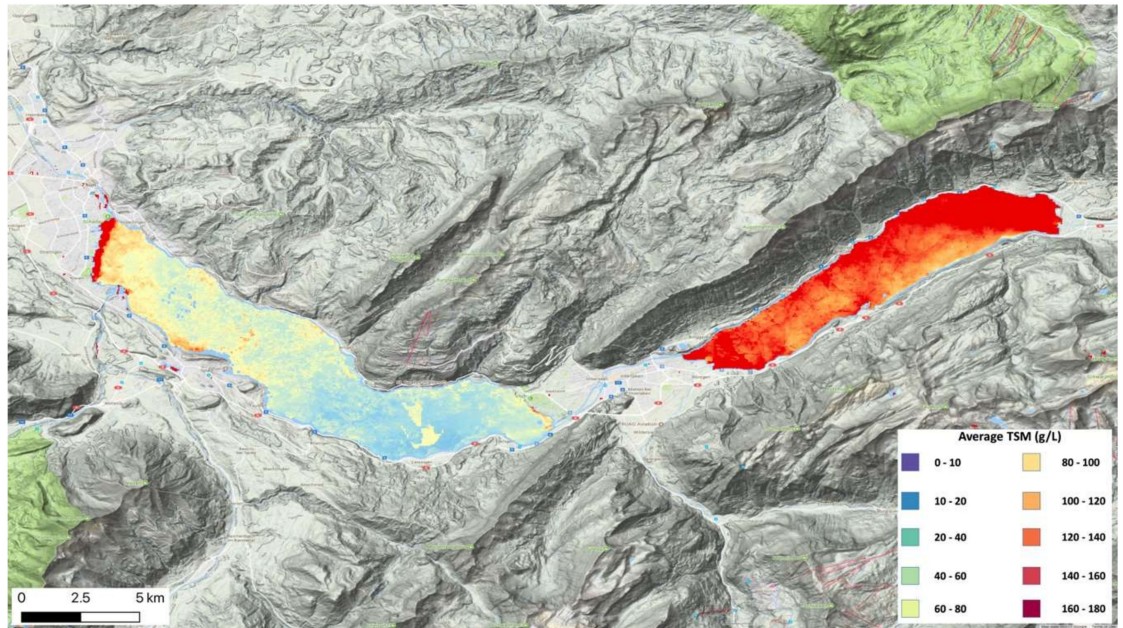

**Figure 6.** Total suspended matter (TSM) concentration for the lakes Thun (**left**) and Brienz (**right**) showing an important difference in terms of sediment load. Data sources: Landsat; Google.

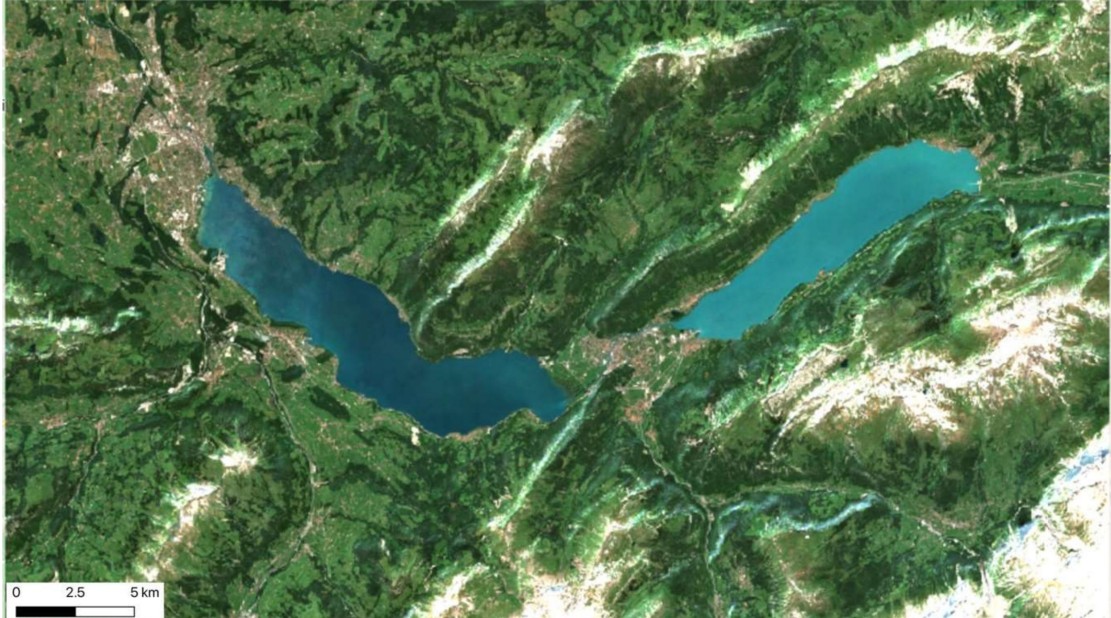

**Figure 7.** True color composite of lakes Thun (**left**) and Brienz (**right**). Brienz lake shows desaturated colors indicating that sediment load is more important than in the Thun lake. Data source: Landsat.

Finally, to derive some EBV, the Normalized Difference Vegetation Index (NDVI) is a well-accepted index for vegetation monitoring. In particular, it can help to inform the seasonal pattern of vegetation activity (i.e., phenology) and can help to evaluate vegetation dynamics and health at different scales [136]. NDVI is a measure of the state of vegetation health based on how the vegetation reflects light in the

red (i.e., strongly absorbed) and near-infrared (NIR) (i.e., strongly reflected) bands. It is calculated as follows [137]:

$$NDVI = (NIR - RED)/(NIR + RED), \tag{2}$$

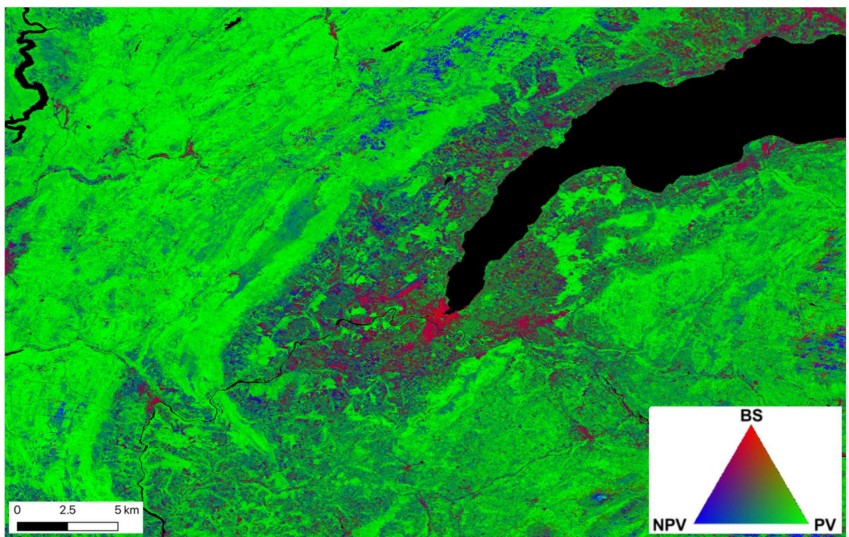

**Figure 8.** Fractional cover in the Great Geneva area using Landsat 7 (Summer 2003).

Index values range from −1.0 to 1.0 representing roughly vegetation for positive values; clouds-water-snow for negative values; and bare soils and rocks for values close to zero [138]. When vegetation is affected by dehydration or becomes sick usually the colors turn from green to brown. This change is particularly significant in NIR light that is increasingly absorbed when vegetation deteriorates. Therefore, NDVI can provide accurate information on the presence of chlorophyll and can be correlated with plant health [139]. In the SDC, we used the script mentioned previously for generating NDWI annual and seasonal means by adapting Formula (1) to compute annual and seasonal NDVI means nation-wide from 1984 to 2019. Vegetation seasonal patterns can be observed as exemplified in Figure 9.

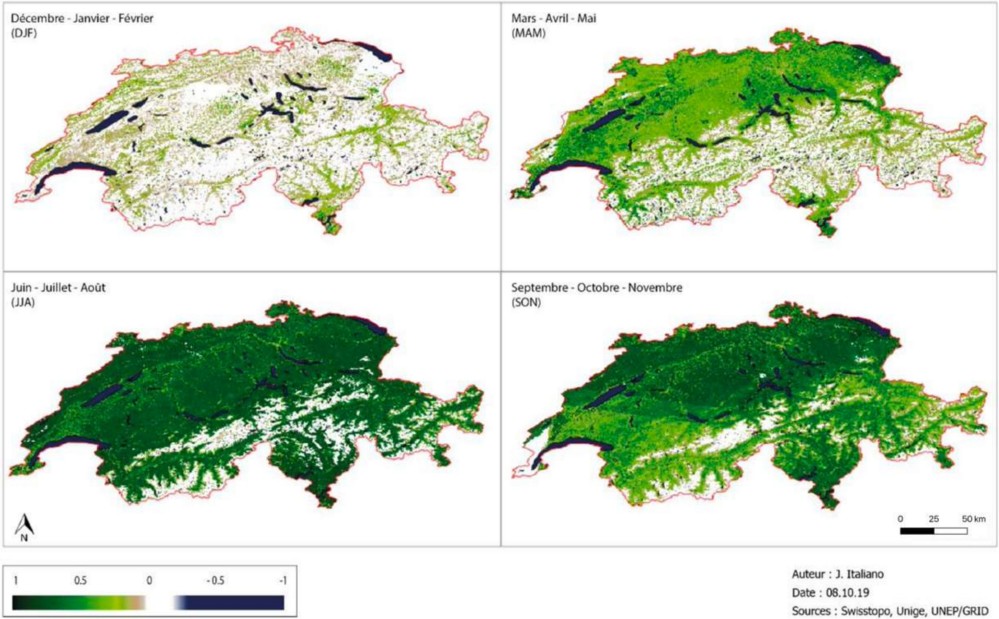

**Figure 9.** NDWI seasonal mean (Year 2019) from Winter (**upper-left**), Spring (**upper-right**), Summer (**lower-left**) and Autumn (**lower-right**). These maps show the evolution over a year of the water content in vegetation and soils.

Similar to the NDWI example, zonal statistics can be computed for the same protected area (i.e., Bois de Finges) annual mean time-series have been computed showing an increase of the NDVI measurements over time (Figure 10). One significant hypothesis that can be made is that conditions for vegetation growth have improved (i.e., more precipitation and heat) caused by climate change at mid-latitudes and, therefore, vegetation can be observed to be more active. Similar patterns have been observed in different parts of the World [140–142] and further studies are needed to confirm and characterize this change across the different biogeographical regions and landscapes of Switzerland.

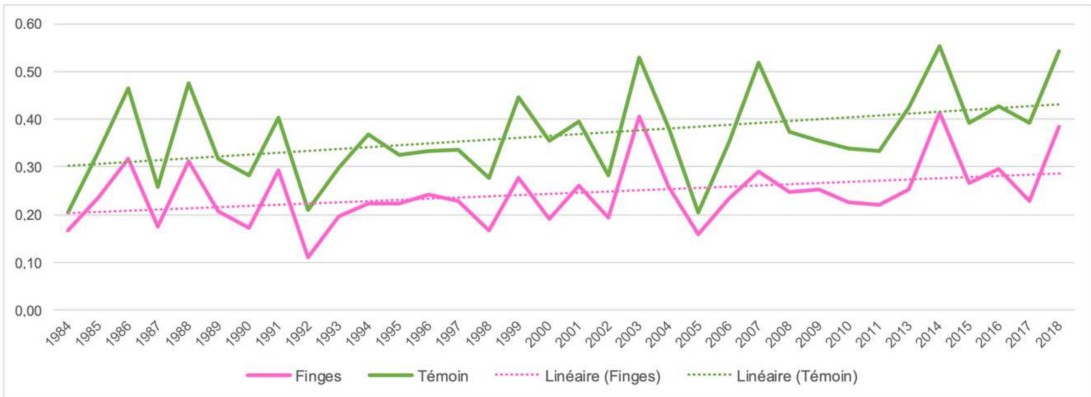

**Figure 10.** Normalized Difference Vegetation Index (NDVI) Annual mean time-series for the Bois de Finges protected area showing an important increase of the vegetation's greenness over the period considered.

## 4. Discussion

The presented use-cases are, to our knowledge, among the first attempts to compute various essential variables to support environmental monitoring using an Earth Observations Data Cube. The first results indicate that EODC can facilitate the production of EVs and can make them accessible to different users. In particular, this offers a consistent analytical framework for space and time-series analysis using satellite EO data. Initial implementation indicates benefits, limitations and further development to strengthen the efficient and effective generation of EVs for environmental monitoring.

In terms of benefits, EODC can help to improve the accessibility, processing and use of EVs [18] providing access to analysis-ready data. It can therefore facilitate the generation of information and knowledge emphasizing the importance of suitable connections between data, applications and users [31,143]. Importantly, such technology can help to move away from traditional data-centric approaches and can enhance more user-/knowledge-centric approaches by lowering the barriers to use of satellite EO data [8] and automating the extraction of actionable information and knowledge at different spatial and temporal scales [144]. This is an essential condition to efficiently support public policy decision-making. Ultimately, it can allow the provision of decision-ready products that can be used for effective decision-making and evidence-based policy-making [14]. More importantly, it allows quantifying measurements and can provide a unique capability to track environmental changes across a country. Finally, EVs can contribute to support the production of high-level environmental indicators defined in different policy frameworks such as the Sustainable Development Goals (SDGs) [4,110], the Aichi targets on Biodiversity [145], the Water-Food-Energy Nexus [146] as well as the Driver-Pressure-State-Impact-Response framework [29].

Other identified benefits are: (1) scalability and reproducibility of code/methods at different scales [37,147] that can be applied in more than 100 countries that are using the Open Data Cube software; (2) integration of the information products into national environmental information database; (3) support innovation and growth in the digital economy; (4) improve efficiency and effectiveness of government investments; (5) improve management of natural resources and (6) stimulate research.

The major limitation is probably the lack of awareness about the EV concept both in some scientific domains as well as in the policy context [15,24]. In addition, EODC is a relatively new technology requiring new knowledge and capacities (e.g., Big Data, artificial intelligence, data science). Therefore, to reach wide adoption, acceptance and commitment, capacity development efforts are fundamental [148]. In that sense, GEO can play a fundamental role in leveraging the potential of EODC and EVs. In its 2020–2022 work programme (http://www.earthobservations.org/geoss_wp.php) there are related activities both on capacity development and EVs. Therefore, GEO can help advocating and strengthening the role of EO and EVs and their associated benefits to society, enabling different stakeholders to become familiar with this technology and raise awareness on the potential of EVs for enhanced environmental monitoring. Benefits can be demonstrated through appropriate examples, best practices, and guidelines. Particular attention should be devoted to developing capacities at the following three levels [149]: (1) human (e.g., education and training of individuals); (2) institutional (e.g., enhancing the understanding within organization and governments of the value of EO data to support decision-making); and (3) infrastructure (e.g., installing/configuring/managing the needed technology). Ultimately, all these efforts should be geared towards a more open and reproducible EO science [37], an approach of research and education that aims to enhance scientific accountability and credibility by making scientific research and knowledge (e.g., data, methods, tools and resources) more collaborative, transparent and accessible through digital technologies.

Another limitation that needs to be overcome is that most of the reviewed EVs are not directly generated from remote sensing measurements. Satellite EO data are often a data source that needs to be used in conjunction with either in-situ/crowdsourced measurements or integrated into models to properly generate EVs. Consequently, efforts should be made to better understand how to link EODC with other data sources to develop efficient workflows and produce EVs in a consistent and coherent manner. To tackle this issue, the system of systems (SoS) approach promoted by GEO in developing GEOSS, a platform interlinking existing and planned observation systems, can be an interesting solution [150,151]. The GEOSS platform relies on interoperability arrangements, a set of standardized technical specifications that allows data from different systems to be combined into coherent data sets [152]. Recent efforts have been devoted to implement Open Geospatial Consortium (OGC) and International Organization for Standardization (ISO) specifications in EODC. These standards are widely adopted in the geospatial community to make EO data and algorithms interoperable [153,154]. With the help of these standards this can prevent EODC from becoming silos of information and facilitate the integration of EO data with different data sources such as in-situ measurements or socio-economic data at different scales (e.g., local, national, international). This is a fundamental condition to leverage the information potential of EO data. All these data sources enclose unique insights and spatiotemporal signatures of relevant environmental changes and can provide crucial information to improve understanding, responding and adequately addressing environmental and sustainability challenges from local to global scales [144,155].

These first results are encouraging and prove the feasibility of generating EVs using an EODC. However, further work is needed to routinely produce EVs. First of all, it is necessary to explore how best to use in-situ data and models together with remotely-sensed data provided by EODC to generate other EVs (i.e., those identified as partial contributions in Section 2). Second, it is important to make the information products interoperable [44,153] and compatible with find-accessible-interoperable-reproducible (FAIR) data-sharing principles [156,157] to facilitate their reuse by different users. Third, all these information products can be the basis for developing a national (or other scales) database for environmental monitoring using satellite EO data and add other EVs such as land cover [158,159]. Fourth, it would be valuable to test the presented approach in other emerging EV domains to refine the understanding of the potential of EODC to support additional disciplines.

The examples shown in this paper demonstrate that EODC and EVs have the potential to realize the objective of transforming data into actionable information to support decisions based on evidence and improve the management of natural resources. This can help countries and local decision-makers

to establish and operate their own analytical capabilities to underpin regulatory and official reporting mandates supported by replicable, reusable and scientifically accountable decision-ready products [31].

## 5. Conclusions

Nowadays, petabytes of EO data and geospatial information are freely and openly available from different data holdings. Combined with open source analytical tools for efficiently and effectively storing, managing and analyzing these data are leading an increasing number of countries to use EODC and ARD to derive useful information and new insights on environmental changes to inform policies and support decision-making.

Essential variables can play an important role in providing necessary information on key elements to adequately describe different components of the Earth system. Moreover, they can also significantly contribute to high-level policy frameworks such as the SDGs. Consequently, it is necessary to understand how to generate EVs on a regular basis.

In this paper, we reviewed EVs in the climate, biodiversity and water domains and produced a comprehensive list of EVs that can take advantage of consistent time-series satellite data. Through various examples, we demonstrated the applicability of EODC to derive EVs and how they can be used both at the local and at the national scales for environmental monitoring. The initial implementation has shown that it is technically feasible, enabling the rapid production of different EVs and benefiting from the spatial and temporal dimension of satellite EO data. However, further work is needed to better understand how to derive EVs using in-situ and or modeled data used in combination with satellite data. We practically demonstrate the objectives of EV to have a high impact, high feasibility and relatively low cost of implementation and such work can be a contribution to the new GEO Initiative on Essential Variables (http://earthobservations.org/documents/gwp20_22/geo_essential_variables_ip.pdf).

To conclude, we think that EODC represents a valuable opportunity to strengthen environmental monitoring systems, facilitating the production of essential information in a consistent and comparable manner to support evidence-based policy-making.

**Author Contributions:** Conceptualization, G.G., E.E. and B.C.; methodology, G.G., E.E. and B.C.; validation, E.E. and B.C.; formal analysis, E.E., J.I., C.P.; investigation, G.G.; data curation, B.C.; writing—original draft preparation, G.G., E.E., J.I., C.P., J.-P.R. and B.C.; writing—review and editing, G.G., E.E., J.-P.R. and B.C.; supervision, G.G.; project administration, G.G.; funding acquisition, G.G. All authors have read and agreed to the published version of the manuscript.

**Funding:** This research was funded by European Commission "Horizon 2020 Program" ERA-PLANET/GEOEssential project, grant number 689443. Results of this publication are partly or fully relying on the Swiss Data Cube (http://www.swissdatacube.org), operated and maintained by UN Environment/GRID-Geneva, the University of Geneva, the University of Zurich and the Swiss Federal Institute for Forest, Snow and Landscape Research WSL.

**Acknowledgments:** The authors would like to thank the Swiss Federal Office for the Environment (FOEN) for their financial support to the Swiss Data Cube. The views expressed in the paper are those of the authors and do not necessarily reflect the views of the institutions they belong to.

**Conflicts of Interest:** The authors declare no conflict of interest.

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
