# Peer review of "Essential Variables for Environmental Monitoring: What Are the Possible Contributions of Earth Observation Data Cubes?"

_data, 2020_

Round 1

Reviewer 1 Report

The manuscript presents an important application of using earth observation data cubes for environmental monitoring.

The text is well written. Nevertheless I have a few comments:

  1. The temporal and spatial resolution of earth observation data is not clearly addressed. The tables in section 2 only show if complete or partial derivation of EV's are possible with earth observation, but the manuscript lacks a discussion about the appropriate spatio-temporal scales for monitoring at different levels of decision making.
  2. Although earth observation data cubes are very promising, you indicate that combination with other data (e.g. in situ) has still to be studied/developed. Other data networks (e.g. GEOSS) are investing a lot in infrastructure. I would like to see a more coherent view on how data cubes fit in the landscape of other (SDI) infrastructures at international and national level and a vision on how local data can be integrated. This can be elaborated for example around lines 367-368 where this is briefly mentioned.
  3. It is great that we move from data-centric to more knowledge centric. You indicate in lines 357-363 that the major limitation are awareness of the EV concept and the requirement of new knowledge and capacities. The latter needs some more elaboration, because it is essential for the uptake. Which skills need to be developed that are now not yet included in university curricula? Which investments need to be done?

Minor comments:

  • In the abstract the abbreviations need to be introduced in brackets after mentioning the full for the first time.
  • Lines 50-53: Improve the phrasing
  • Lines 70-73: Add here the lack of capacity/skills
  • Figure 1: add % to legend, add scale bar, add data source to map or caption
  • Figure 2: mention the year in the caption. Indicate the location of Bois de Finges that is discussed further. Use a scale bar instead of scale text (which will probably be wrong in print).
  • Figure 4: legend is too small. 1 legend is sufficient for both maps. Source needs to be added.
  • Figure 5: 1 legend is sufficient for both maps. Source needs to be added.
  • Figure 6 and 7: scale bar needs to be added. Source needs to be added.
  • Figure 9: use scale bar instead of scale text (see before).
  • Line 348: Change "framework" to "frameworks"

Author Response

We thank Reviewer 1 for his valuable comments. All our answers are detailed in the attached Word document.

Reviewer 2 Report

This paper reports a data cube concept and a system (maybe) implemented from it to extract a variety of environmental variables from managed data produced from LANDSAT and Sentinel series of satellites. The research is interesting and especially under a spatiotemporal aspects to manage large historical data in a spatiotemporal data analytical fashion. Would suggest a few aspects to improve: 

  1. describe the methodology in detail, especially: 
    1. how was the data cube implemented
    2. how the framework/architecture was developed
    3. what software was used to conducted the data/information subsetting/extraction
  2. list all data available for users to freely download among all the environmental variables or how the results are eventually made available to end users. 
  3. Check the references, there's a lot but make sure no duplication of sequence numbers and formats are in compliance
  4. have a native speaker check the language and detail expression. 
  5. The following references maybe of interest to justify the research and state the importance of the data framework from big spatiotemporal data analytical aspects: 

Yang, C., Clarke, K., Shekhar, S. and Tao, C.V., 2020. Big Spatiotemporal Data Analytics: a research and innovation frontier, International Journal of Geographical Information Science, 34:6, 1075-1088

Yang, C., Yu, M., Li, Y., Hu, F., Jiang, Y., Liu, Q., Sha, D., Xu, M. and Gu, J., 2019. Big Earth data analytics: A survey. Big Earth Data3(2), pp.83-107.

Author Response

We thank Reviewer 2 for his valuable comments. All our answers are detailed in the attached Word document.

Round 2

Reviewer 2 Report

All my comments are addressed, would suggest to accept.